# Exploring Complexity Measures for Analysis of Solar Wind Structures and Streams

Venla Koikkalainen[1], Emilia Kilpua[1], Simon Good[1], and Adnane Osmane[1]

[1]Department of Physics, University of Helsinki, P.O. Box 64, Helsinki, FI-00014, Finland

**Correspondence:** Venla Koikkalainen (venla.koikkalainen@helsinki.fi)

**Abstract.** In this paper we use statistical complexity and information theory metrics to study structure within solar wind time series. We explore this using entropy-complexity and information planes, where the measure for entropy is formed using either permutation entropy or the degree distribution of a horizontal visibility graph (HVG). The entropy is then compared to the Jensen complexity (Jensen-Shannon complexity plane) and Fisher information measure (Fisher-Shannon information plane), formed both from permutations and the HVG approach. Additionally we characterise the solar wind time series by studying the properties of the HVG degree distribution. Four types of solar wind intervals have been analysed, namely fast streams, slow streams, magnetic clouds and sheath regions, all of which have distinct origins and interplanetary characteristics. Our results show that, overall, different metrics give similar results but Fisher-Shannon, which gives a more local measure of complexity, leads to a larger spread of values in the entropy-complexity plane. Magnetic cloud intervals stood out in all approaches, in particular when analysing the magnetic field magnitude. Differences between solar wind types (except for magnetic clouds) were typically more distinct for larger time lags, suggesting universality in fluctuations for small scales. The fluctuations within the solar wind time series were generally found to be stochastic, in agreement with previous studies. The use of information theory tools in the analysis of solar wind time series can help to identify structures and provide insight into their origin and formation.

## 1 Introduction

The solar wind is permeated by multi-scale fluctuations in its magnetic field and plasma parameters (Verscharen et al., 2019; Bruno and Carbone, 2013). These fluctuations play a pivotal role in shaping the evolution and dynamics of the solar wind, driving heliospheric turbulence and facilitating energy transfer across scales. Furthermore, solar wind fluctuations contribute to the transport and acceleration of charged particles, and can strengthen the coupling between the solar wind and planetary magnetospheres, therefore leading to stronger space weather effects (Oughton and Engelbrecht, 2021; Borovsky and Funsten, 2003; Osmane et al., 2015; Telloni et al., 2021; Kilpua et al., 2017b).

The solar wind exhibits large-scale structure, characterized by alternating fast ($\gtrsim 600$ km/s) streams coming primarily from coronal holes, and slower ($\sim 300-400$ km/s) streams with variable sources, such as release of initially confined plasma from the streamer belt region or outflows from the edges of coronal holes (Zirker, 1977; McComas et al., 2003; Cranmer, 2009; Brooks et al., 2015; Bale et al., 2019). Interaction between streams of different speeds form compressive structures known as stream

interaction regions (SIRs; Richardson, 2018), which often repeat in 27-day intervals as the coronal holes from which they originate are relatively long-lived structures. Another key category of large-scale heliospheric structures are the interplanetary counterparts of coronal mass ejections (ICMEs; Kilpua et al., 2017a). During solar maximum, ICMEs can comprise up to 40-60% of the ecliptic solar wind near the Earth's orbit (Richardson and Cane, 2012). A typical ICME in the solar wind consists of a leading shock wave, a turbulent sheath region and an ejecta, provided that the ejecta propagates sufficiently fast with respect to the preceding solar wind. Approximately one-third of the ICME ejecta shows signatures consistent with an underlying flux rope configuration (Richardson and Cane, 2004), i.e., enhanced magnetic field magnitude, smooth rotation of the magnetic field direction over a large angle, and depressed proton beta. Such events are commonly referred to as magnetic clouds (MCs) (Burlaga et al., 1981). In contrast to MCs, ICME sheaths, being compressive structures, are more similar to SIRs in their solar wind properties, exhibiting large-amplitude magnetic field variations and relatively high densities and temperatures.

Due to their distinct origins and formation, the various types of solar wind discussed previously (slow, fast, SIRs, sheath and ejecta) are expected to feature significant differences in their fluctuation and turbulence characteristics (Kilpua et al., 2017b). Several studies have already indicated that normalized magnetic field fluctuations are higher during the compressive sheaths and SIRs than in the unperturbed solar wind, while magnetic clouds represent the lowest fluctuation levels (Kilpua et al., 2017a; Borovsky et al., 2019; Moissard et al., 2019).

Fluctuations in large-scale solar wind structures can arise from multiple sources. In the fast wind, the most common fluctuations are anti-sunward propagating Alfvén waves, believed to originate from the convective motions of the solar photosphere (Belcher and Davis, 1971). In the corona, sunward propagating Alfvén waves are generated from reflected outward waves and via parametric decay instability (Shoda and Yokoyama, 2016; Tenerani and Velli, 2013; Sishtla et al., 2022), leading to an active turbulent cascade of energy from large to smaller scales. There is evidence that turbulence is also actively generated further out in the heliosphere, suggesting that inward Alfvén waves must also be generated in the heliosphere (Chen et al., 2020). Moreover, some of the fluctuations in the solar wind arise from intermittent coherent structures that may be unrelated to the turbulent cascade. Examples include current sheets, flux tubes, and small-scale flux ropes, which may originate either from the Sun or be created in interplanetary space via magnetic reconnection (Borovsky, 2008; Li et al., 2011; Sanchez-Diaz et al., 2017; Zhao et al., 2021; Ruohotie et al., 2022).

An important question regarding solar wind fluctuations is whether they are stochastic, periodic or chaotic in nature. This distinction can provide insights into the origin of the fluctuations and mechanisms that generated them. Understanding the nature of solar wind fluctuations is also an important aspect for space weather applications, for example with regard to building better numerical models and forecasting schemes, given that stochastic (random) fluctuations are difficult to predict.

In this study we apply complexity analysis to study fluctuations in the solar wind, which offers a complementary approach to more traditional analysis techniques. Using complexity analysis we can explore phenomena such as cross-scale effects, emergence, and self-organising behaviour (McGranaghan, 2024). This is particularly relevant to the study of the solar wind, where a plethora of fundamental plasma processes are in action. These processes cause structures from small-scale turbulent fluctuations to large-scale phenomena such as ICMEs. While complexity science or information theory may not directly explain the underlying physical processes of the analysed systems, they can provide valuable insights into patterns and structures in

solar wind time series, help to identify the combined effects of interacting subsystems, and differentiate between solar wind structures of different origin (Kilpua et al., 2024). Our aim is to explore techniques that are new to solar wind studies (HVG analysis and the Fisher-Shannon information plane) in combination with a technique that has been used previously in the field, Jensen-Shannon complexity. These methods, which will be introduced in the next paragraphs, are complementary to each other.

The Jensen-Shannon complexity analysis (Rosso et al., 2007) has recently become more widely used in the field of space plasma physics. It is based on the concept of permutation entropy, i.e., on finding how different permutation patterns occur in a time series (Bandt and Pompe, 2002). Permutations at different time lags can be determined for the fluctuations, straightforwardly allowing for a multi-scale analysis. Previous studies using the Jensen-Shannon entropy have found solar wind magnetic field fluctuations to be stochastic in nature (Weck et al., 2015; Weygand and Kivelson, 2019; Good et al., 2020; Kilpua et al.,

2022; Raath et al., 2022; Kilpua et al., 2024). In particular, Kilpua et al. (2024) performed an extensive permutation entropy and complexity analysis study of different types of solar wind using 1 au measurements for the period 1997-2022. They found that at large scales (i.e. fluctuations at time lags of a few minutes), magnetic clouds clearly exhibited the lowest entropies and highest Jensen-Shannon complexities, while fast wind streams were the most stochastic. At smaller scales, turbulent features were more similar. In their analysis of fractal dimensions, Muñoz et al. (2018) found that magnetic clouds similarly stood

out from other solar wind types, with the clouds displaying a distinctive monofractal behaviour. Macek (2010) investigated the fractal nature of the solar wind, and argued that solar wind fluctuations showed signatures of low-dimensional attractor. However, the studies using Jensen-Shannon complexity have thus far found no signatures of low dimensional attractor structure within solar wind.

The Jensen-Shannon complexity analysis is one of a number of methods to investigate the nature of fluctuations. Others

include the visibility graph (VG) (Lacasa et al., 2008), a method that transforms the analysed time series into a graph that permits investigation of underlying patterns and estimation of complexity. The method is based on determining whether two values in the time series are 'visible', i.e. connected. A special case of the VG method is the horizontal visibility graph (HVG) (Luque et al., 2009), where connections are made based on a more simple rule for 'visibility' than in the traditional VG. The HVG technique can be used in combination with the Fisher information measure (FIM; Fisher, 1925; Ravetti et al., 2014) to

study the complexity of a time series. The FIM is a more local measure than Shannon entropy as it compares consecutive values to each other. Thus, the ordering of the distribution is important in Fisher's approach. FIM has previously been used in the field of heliophysics by Balasis et al. (2016, 2023b) who used entropy and Fisher information to study geomagnetic jerks and geomagentic activity indices, respectively.

Other methods that have been used in heliophysics are, for example, the recurrence quantification analysis (Donner et al.,

2019), network analysis (Orr et al., 2021), and maximal Lyapunov exponents, approximate entropy analysis, and delay vector variance (Oludehinwa et al., 2021). A comprehensive review of complexity science approaches and techniques in heliophysics is provided by McGranaghan (2024), while Balasis et al. (2023a) focuses specifically on complex methods and their usage in the Near-Earth environment. Finally, Chian et al. (2022) review nonlinear dynamics and plasma turbulence, expanding on the concepts also discussed in this study, such as chaotic and stochastic dynamics and complexity.

The key purpose of this analysis is to examine how the Jensen-Shannon complexity, the Fisher-Shannon information plane, and HVG-analysis capture the fluctuation signatures of distinct solar wind structures. It can be expected that different types of solar wind could show different types of fluctuations perhaps relating to the processes that cause their formation. We thus analyse in detail a few selected events that are representative of four large-scale solar wind categories, namely slow wind, fast wind, magnetic clouds and sheaths. The paper is organised as follows: in Section 2 we present the data and analysis methods, including detailed descriptions of each of the complexity measures used; in Section 3 the results are presented; and in Sections 4 and 5 we discuss and conclude.

## 2    Data and Methods

### 2.1    Spacecraft data

The solar wind data used in this study comes from the Wind spacecraft. We used the 3-second resolution data from the Magnetic Field Investigation (MFI) instrument, which is a boom-mounted dual triaxial fluxgate magnetometer (Lepping et al., 1995). Measurements are given in geocentric solar ecliptic (GSE) coordinates. Three intervals of data were considered for each solar wind type, each consisting of 12 hours of measurements. For fast wind intervals were 1) 28.12.2005 00:39-12.38 UT, 2) 9.4.2006 22:33 UT - 10.4.2006 10:32 UT, 3) 14.3.2007 4.23-16.22 UT. For slow wind: 1) 26.12.2005 14:11 UT - 27.12.2005 2:10 UT, 2) 8.4.2006 7:21-19.20 UT, 3) 10.3.2007 17.05 UT - 11.3.2007 5:04 UT. For sheath regions: 1) 17.9.2011 3:02-15:01 UT, 2) 26.2.2012 21:04 UT - 27.2.2012 9:03 UT, 3) 27.6.2013 13.56 UT - 28.6.2013 1:55 UT. For MCs: 1) 15.5.2005 10:00-21:59 UT, 2) 20.5.2005 18:00 UT - 21.5.2005 5:59 UT, 3) 12.6.2005 22:00 UT 13.6.2005 9:59 UT. These intervals were chosen from the data set of Kilpua et al. (2024), with the requirement that there should be as few data gaps as possible in the data to robustly calculate the complexity measures. All of the time series we analysed had less than 2.4% of missing data points. Time series plots of the magnetic field magnitudes and components during these intervals are included in the Appendix, Figures A1, A2, A3 and A4.

Due to the differences between the various techniques used to estimate the complexity, we have applied two different approaches to account for data gaps in the time series. For the HVG approach, data was concatenated into a final time series such that all data gaps were closed. When calculating the permutation entropy, we followed the suggestion presented in Olivier et al. (2019), i.e., we excluded all those permutation patterns (of the chosen length) from the calculation of the permutation entropy that contained missing data. This resulted in excluding less than 3 percent of the intervals in all cases except for the third sheath region, and the first magnetic cloud, where 6.3 and 12.2 percent were excluded, respectively.

### 2.2    Horizontal visibility graphs

The visibility graph (VG) was first introduced by Lacasa et al. (2008) as a way of converting time series into networks. The method is based on studying the 'visibility' of data points to each other, i.e. the amplitude of values in a time series. Two data points of small magnitude separated by a high magnitude data point do not 'see' each other and are hence not connected,

while two large magnitude data points separated by many lower magnitude data points are visible to each other. Each point in the original time series corresponds to a node in the resulting graph. Studying time series in network form enables the use of network analysis methods that are powerful tools to assess the nature of time series; i.e., whether the processes that create them are chaotic, periodic or stochastic in nature. The horizontal visibility graph (HVG) is a simplification of the VG, introduced by Luque et al. (2009). In the HVG, connections between data points (i.e. nodes) are made based on how different points in the data set are 'visible' to each other in the horizontal direction. Two points $x_a$ and $x_b$ in a time series will be connected (i.e. visible) if:

$$x_a, x_b > x_n \tag{1}$$

for all $n$ such that

$$a < n < b. \tag{2}$$

Each point in the series will be connected at least to its two neighbours, which makes the resulting graph fully connected. The connections in the map can be studied statistically by forming a degree distribution of the graph. The degree ($k$) of a node measures how many connections a given node has with the other nodes in the series. For example, if a data point/node is only connected to its neighbours, its degree will be two. The degree distribution therefore gives the number of nodes that have $k$ connections for the range of possible $k$ values.

As the relation for connecting the points in the investigated time series is rather simple, some analytical solutions can be derived to describe the properties of the final graph if considering purely uncorrelated (i.e. random walk) time series data (Luque et al., 2009), an example of which being regular Brownian motion. Lacasa and Toral (2010) tested the method for uncorrelated as well as for chaotic and stochastic data with correlation, fractional Brownian motion being one example. They found that all of the tested time series (i.e. not only uncorrelated data) follow an exponential equation for $P(k)$ in the tail of the degree distribution: $P(k) \sim \exp(-\lambda k)$. Lacasa and Toral (2010) then classified some known chaotic and stochastic processes by forming HVGs and determining the $\lambda$ values of the distributions. The $\lambda$ value can be estimated by the exponential fitting to the degree distribution. The authors show that $\lambda = \ln(3/2)$ is the threshold between chaotic and stochastic processes. The values $\lambda < \ln(3/2)$ correspond to chaotic processes and $\lambda > \ln(3/2)$ to correlated stochastic processes. When $\lambda \sim \ln(3/2)$ the process is uncorrelated.

The above described classification method by Lacasa and Toral (2010) was tested with a more comprehensive set of chaotic and stochastic processes by Ravetti et al. (2014). The authors identified several issues with the method. For example, the fractional Gaussian noise, which is a stochastic process, was incorrectly classified by the method in some cases. Another issue identified by Ravetti et al. (2014) was that the obtained degree distributions did not always exhibit exponential regions where the fitting could be performed. It should be also noted that even in cases where the exponential region does exist, it is not always clear what $k$-range should be fitted. The fitting is expected to be applied to the tail of the distribution, but, as discussed by Ravetti et al. (2014), it is ambiguous as to what degree the fitting should be started from. In any case, the part of the

distribution at low degree numbers is excluded from the fitting, and as such some information about the connections within the graph is not utilized.

## 2.3 Shannon entropy

Shannon entropy, first proposed by Shannon (1948), is a measure of the information that can be gathered from a set of data. For any discrete probability distribution, the Shannon entropy is given by:

$$S[P] = -\sum_{j=1}^{N} p_j \cdot \ln(p_j), \tag{3}$$

where $P = \{p_j; j = 1, ..., N\}$ is a discrete probability distribution with $N$ possible states. In the case when Shannon entropy is zero it is possible to predict with certainty which of the possible outcomes $j$ will take place. Conversely, the maximal entropy is achieved by a set of data where it is very difficult to predict the outcome, i.e the probability distribution is uniform or close to uniform. The maximal entropy for a system can be used to normalize the Shannon entropy (Martin et al., 2006):

$$H[P] = S[P]/S_{max}$$

The probability distribution function (PDF) used in calculating the Shannon entropy can be formed in several ways.

## 2.4 Permutation entropy

Bandt and Pompe (2002) define a measure of complexity, permutation entropy, based on studying neighbouring values in a time series. The approach is similar to Shannon entropy but with a specific PDF that is based on the magnitudes of points in a time series. Permutation entropy for a series $\{x_t\}_{t=1,..N}$ is defined as:

$$S(P) = -\sum_{i=1}^{d!} p_i \log_2 p_i, \tag{4}$$

where $P$ is the probability distribution of the patterns found in the time series, $p_i$ is the probability of a pattern where $i = 1, 2, 3, ..., d!$, and $d$ is the embedding dimension, i.e. the length of the subset where the permutations are found. The normalized entropy is defined as:

$$H(P) = -S(P)/\log_2 d! \tag{5}$$

and the PDF, i.e. the probability of each permutation (amplitude ordering) for a time series of length N is:

$$p(\pi) = \frac{\#\{t \mid t \le N - d, (x_{t+1}, ...x_{t+n}) \text{has type } \pi\}}{N - d + 1} \tag{6}$$

where $\#$ indicates number.

The form of the permutation entropy equation is the same as the Shannon entropy, but here the PDF is specifically the permutation distribution. In the context of this study, permutation refers to the relative ordering of the data points in the selected sample from the investigated time series.

The analysis is restricted by the choice of embedded dimension $d$, which is the length of the subsection of the time series that is studied, i.e. the number of data points in a sample. Additionally, a time lag $\tau$ can be applied. This results in choosing every $n$th value of the series instead of consecutive values when forming the patterns. Olivier et al. (2019) found that $H$ values become stable when choosing a $\tau$ value of $\sim 20$ or higher for the data where averaging has been applied.

## 2.5 Jensen-Shannon plane

Martin et al. (2006) introduce the Jensen-Shannon statistical complexity, which can be used in combination with Shannon entropy:

$$C_{js} = -2\frac{S\left(\frac{P+P_e}{2}\right) - \frac{1}{2}S(P) - \frac{1}{2}S(P_e)}{\frac{d!+1}{d!}\ln(d!+1) - 2\ln(2d!) + \ln(d!)}H(P) \tag{7}$$

where $P_e$ is the maximum permutation entropy.

Jensen-Shannon complexity is a measure of order in a system. It indicates how different the probability distribution is from a uniform distribution for a given value of normalized entropy $H$. The value of $C_{js}$ is the largest for the case where the distribution is most varied and the smallest both for high order and high disorder.

The 'complexity plane' refers to a plane where Jensen-Shannon complexity $C_{js}$ is plotted against normalized permutation entropy $H$. It was first introduced by Rosso et al. (2007), who showed that chaotic, stochastic and periodic series fall in different areas of the plane. There are also clearly defined maximum and minimum values of complexity for each entropy value that correspond to disorder and perfect order (Martin et al., 2006). The robustness of the analysis can be evaluated with the following tests: $N/d! > 10$ and $\sqrt{d!/N - (d-1)r} < 0.2$ (Weygand and Kivelson, 2019; Osmane et al., 2019), where $r$ is the sub-sampling rate, which relates to the time lag: $\tau = r\Delta t$, where $\Delta t$ is the data resolution.

## 2.6 Fisher-Shannon plane

Fisher's information measure (FIM), first introduced by Fisher (1925), is another parameter to estimate the nature of time series and embedded structures. Compared to Shannon entropy, FIM is more of a local measure. In FIM, consecutive values of the distribution are compared to each other, while the Shannon entropy is a measure of the full probability distribution (Ravetti et al., 2014)).

In this study, the discrete form for FIM, following Gonçalves et al. (2016), will be used:

$$\text{FIM} = F_0 \sum_{i=1}^{N-1} \left((p_{i+1})^{1/2} - (p_i)^{1/2}\right)^2 \tag{8}$$

where the constant $F_0$ is:

$$F_0 = \begin{cases} 1 & p_{i^*} = 1 \text{ for } i^* = 1 \text{ or if } i^* = N \text{ and } p_i = 0 \ \forall \ i \neq i^* \\ 1/2 & \text{otherwise} \end{cases} \tag{9}$$

Frieden and Soffer (1995) expand on the FIM and its uses in physics. They relate the FIM to the gradient of the the distribution it is applied to, and write that for a uniform PDF the FIM will be small. Such a distribution will describe a highly

unpredictable system. Conversely, for a highly predictable system the PDF will have higher gradient and larger FIM. For example, in the case of a delta distribution, FIM is equal to 1, as the probability is zero everywhere except at $x = 0$. Conversely, for a uniform distribution FIM is zero, as there is no gradient in the distribution. Unlike in Shannon entropy, the ordering of the distribution is important for FIM, as it takes into account adjacent values in the PDF.

Plotting FIM and Shannon entropy on a plane gives the Fisher-Shannon information plane (Vignat and Bercher, 2003).

This plane has been studied, among others, by Olivares et al. (2012a). The authors plotted FIM and Shannon entropy of the investigated data in a Fisher-Shannon plane and found that chaotic and noisy stochastic data fall into different regions. They calculated FIM and entropy using the Bandt and Pompe (2002) permutation distribution as the PDF.

The effect of ordering of the patterns on the Fisher-Shannon plane when calculating FIM from the permutation distribution has been studied by Olivares et al. (2012a), Olivares et al. (2012b), and Spichak et al. (2021). Olivares et al. (2012a) find that

the Lehmer protocol i.e. lexicographic order gives more structure than the other tested pattern, namely the Keller order. In this study we use the lexicographic order for sorting the permutations as it is widely used in various applications.

The Fisher-Shannon information plane can also be formed from the HVG degree distribution as the PDF, as was done by Ravetti et al. (2014) and Gonçalves et al. (2016). Ravetti et al. (2014) introduce this approach as a way of reducing the previously mentioned problems related to the classification using $\lambda$ values. With the Fisher-Shannon plane, the full degree

distribution is taken into account, and no information is therefore left out of the analysis. Ravetti et al. (2014) find that chaotic and stochastic processes fall into different areas on the plane.

## 3   Results

For this study, we formed Jensen-Shannon and Fisher-Shannon complexity/information planes for four types of solar wind time series (Sect. 2.1). For the Jensen-Shannon plane the permutation entropy technique was used to form the PDF, while for

the Fisher-Shannon plane we used both the permutation entropy technique and the HVG degree distributions to form the plane, in order to study the differences between the two ways of forming the probability distribution.

When using permutation entropy to form the Jensen-Shannon and Fisher-Shannon planes, the effect of time lag $\tau$ on permutation entropy was calculated for two subsampling rates $r = 20$ s and $r = 300$ s; with the 3 s cadence data, these correspond to time lags of $\tau = 60$ seconds and 15 minutes, respectively. The embedding dimension $d$ was kept as 5, similar to most previous

studies of the solar wind (see Section 1). Likewise the solar wind results on the complexity/information planes are compared to the fractional Brownian motion (fBm), as in previous studies. The fBm curves were formed by generating 100 samples of fBm noise for nine Hurst exponent values ranging from 0.1 to 0.9, calculating the placements of those series on the planes, and then taking the average of those results to form the final curves that are given in the figures.

Finally, we have tested the $\lambda$-classification proposed by Lacasa and Toral (2010) on the selected solar wind data intervals.

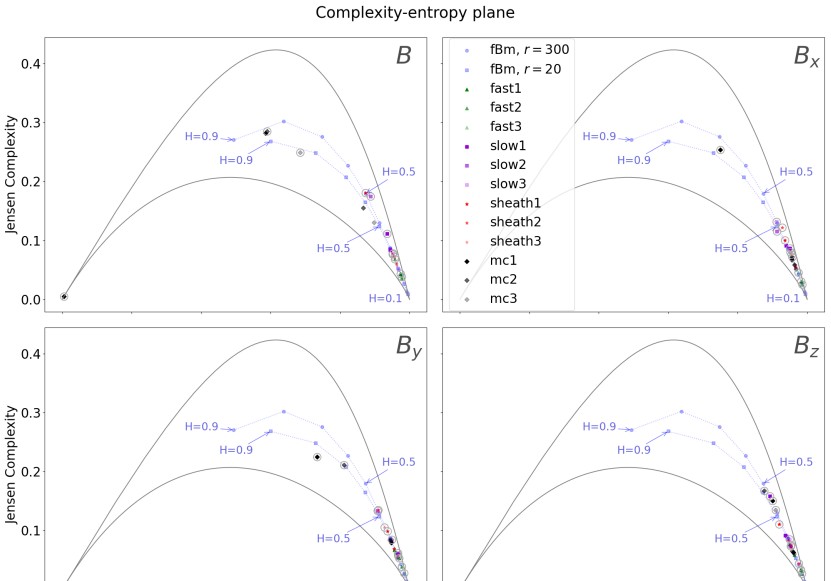

**Figure 1.** The Jensen-Shannon complexity plane showing the placement of data points calculated for different solar wind types (fast, slow, sheath and magnetic cloud). The data points that are calculated for subsampling rate $r = 300$ are surrounded by grey circles and those without the surrounding circle are calculated for $r = 20$. The maximum and minimum curves are given for $r = 300$.

### 3.1 Jensen-Shannon plane

Figure 1 shows the Jensen-Shannon complexity plane for the magnetic field magnitude and the GSE field components. For most of the investigated cases the data points fall onto or close to the fBm curves. Additionally, most of the data points are clustered at the lower right corner of the map, i.e. at the high entropy and low complexity region characteristic of a highly stochastic process. We also note that for most of the studied events there is no significant change in the placement of the data points from $r = 20$ to $r = 300$.

The most notable difference between $r = 20$ and $r = 300$ is found for the magnetic field magnitude, $B$, of magnetic cloud (MC) 1 in the top left corner in Figure 1. The $r = 300$ data point for MC1 is placed at the bottom left corner of the plane, indicating entropy $\sim 0$ and complexity $\sim 0$. For $r = 20$ the same magnetic cloud is placed close to the middle of the plane at entropy $\sim 0.6$ and complexity $\sim 0.3$. Large differences between the $r = 20$ and $r = 300$ data points are present also for MC1 for $B_x$, and both for MC1 and MC2 for $B_y$. In these cases the $r = 300$ data points have considerably larger complexity and smaller entropy than the $r = 20$ data points. The MC1 data points are also the ones that deviate most from the fBm curve. For the magnetic clouds, increasing $r$ results in lower values for entropy and higher values for complexity, with the exception of MC1. We note that complexity is zero both for perfectly ordered and random processes. Most low complexity data points

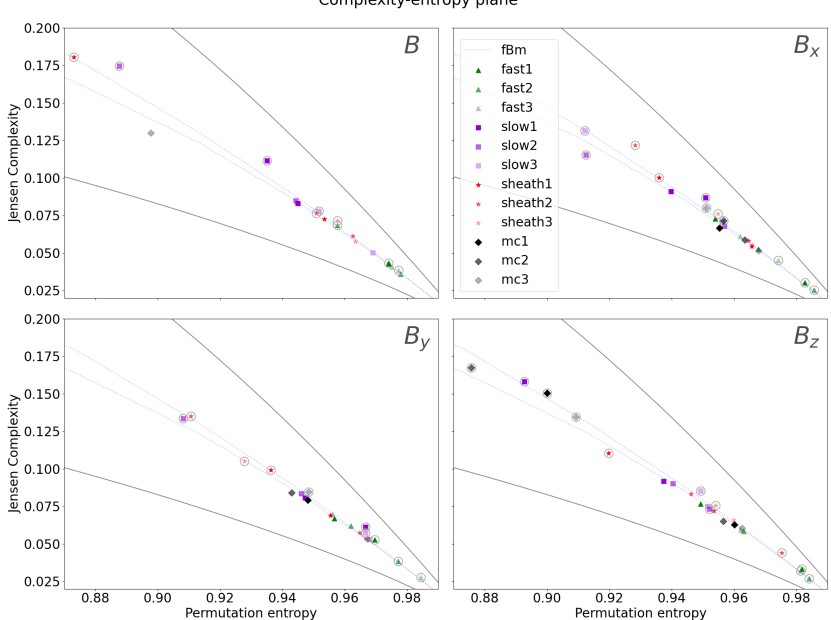

**Figure 2.** A zoom-in to the bottom right corner of the Jensen-Shannon complexity-entropy plane. The symbols and curves are the same as in Figure 1.

shown in Figure 1 are associated with high entropy and likely represent random time series, while the $r = 300$ data point for
MC1 has entropy close zero and thus presents a perfectly ordered time series.

Figure 2 shows a zoom-in at the lower right corner of the complexity-entropy plane. This allows for a more detailed comparison of the ordering of the data points clustered in that region. It is now evident that for the magnetic field magnitude $B$ (top left corner), the fast solar wind has the highest entropy and the lowest complexity. For fast solar wind interval 1 (fast1) the $r = 20$ and $r = 300$ markers overlap. For fast wind interval 3 (fast3) there is also only a very small change between the data points. The fact that entropy and complexity do not change significantly with the sub-sampling rate (i.e. with the time lag) is a signature of a highly stochastic process (Osmane et al., 2019). Next to the fast solar wind on the fBm curve are the sheath regions and slow solar wind. For the fast and slow solar wind and sheath regions, the $r = 300$ data points have lower entropy and higher complexity than the $r = 20$ data points in all cases. This is opposite to what was found for magnetic clouds. The most drastic change between the $r = 20$ and $r = 300$ markers are for the slow wind interval 2 (slow2) and sheath 1. In both of these cases, the $r = 300$ marker has moved considerably up along the fBm curve. We also note that two of the sheath $r = 300$ data points are above the fBm curve. This signifies that they have higher complexity than the fBm process and are associated with more structure.

For the individual magnetic field components, the most distinct finding is that the fast wind again has higher entropies and lower complexities for $r = 300$ than for $r = 20$. For $B_x$ and $\tau = 300$, two of the magnetic clouds (MC1 and MC2) have quite high entropies and low complexities. The sheath regions and slow wind are placed between these magnetic clouds and MC3

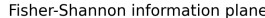

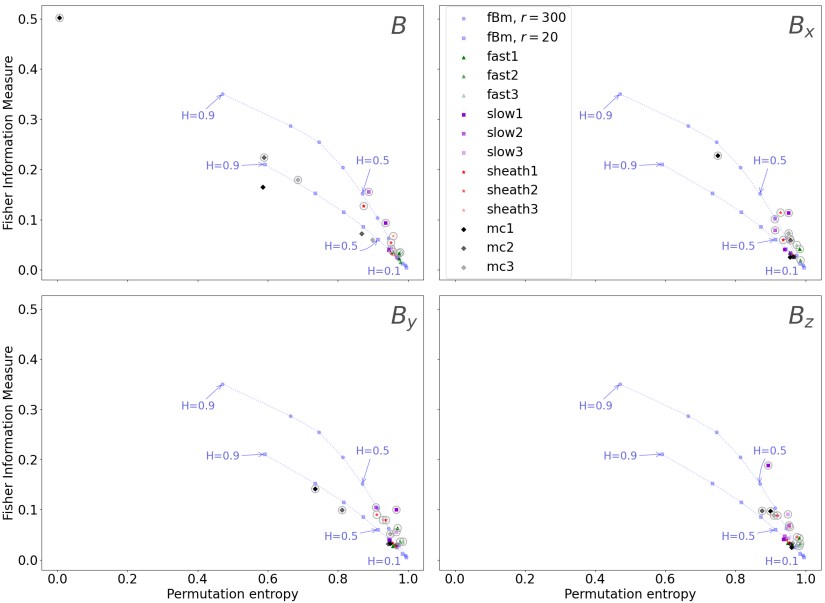

**Figure 3.** The Fisher-Shannon information plane. The solar wind events not marked with a grey circle have $r = 20$, and those marked with the circle are calculated for $r = 300$.

with the lowest entropy that is visible in the figure with the full plane. For $B_y$ the order of the events is approximately similar to previously discussed trends in $B_x$, and for $B_z$, and all the time series have in general higher entropy than is found in the other magnetic field components.

## 3.2 Fisher-Shannon plane

### 3.2.1 Permutation entropy

Next we will investigate the Fisher-Shannon information plane. When permutation entropy is used to calculate the entropy, the horizontal axis is the same as in the previous section for the Jensen-Shannon complexity plane. The vertical axis is now the Fisher Information Measure (FIM). As a result, there are some changes in the vertical placements of the markers on the plane. In general, the data points are spread out more on the plane when using FIM instead of Jensen complexity. This can be seen also in the fBm curves, where increasing the subsampling rate (time lag) results in a considerably larger change in the placement of the curve than on the Jensen-Shannon plane. Similarly to the Jensen-Shannon plane, the MC time series stand out in the FIM plane.

Looking at the magnetic field magnitude $B$, the MCs have lowest entropies and highest FIM values. For $B_x$ and $B_y$ the same MC (MC1) deviates from the cluster of markers at the bottom right corner as in the Jensen-Shannon plane. For $B_z$, one

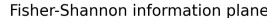

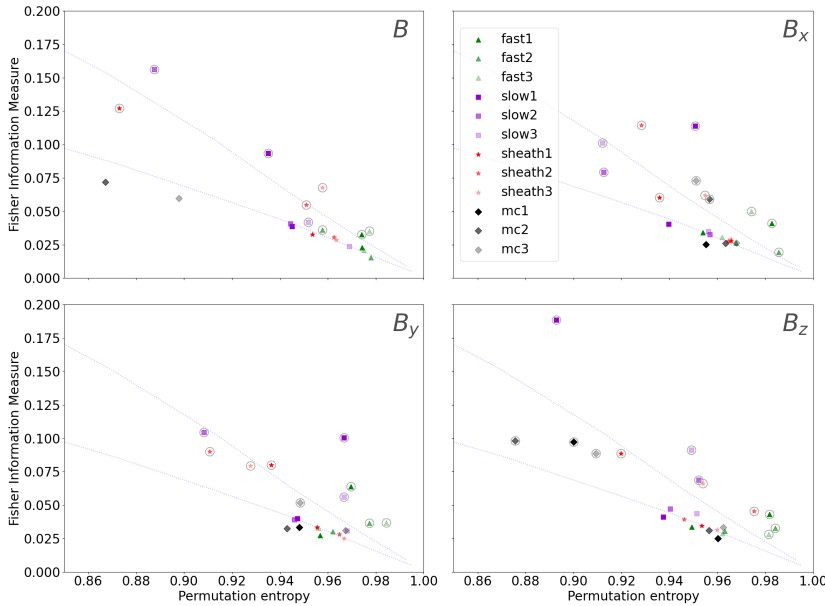

**Figure 4.** A zoomed-in section of the Fisher-Shannon plane. The top dotted curve is the fBm curve with $r = 300$, and the lower curve is the fBm curve with $r = 300$.

of the slow solar wind events (slow1) has clearly the highest FIM, standing out from the other events and significantly away from the fBm curve.

Figure 4 shows a zoomed-in section of the bottom right hand corner of the plane and the spread of the markers can be seen more clearly. For $B$, the fast solar wind markers are again clustered close to the corner of the plane both for $r = 20$ and $r = 300$. Increasing $r$ appears to increase the FIM values for the events. For the magnetic field components, the markers are clustered along the fBm line with no apparent order for $r = 20$, but increasing the subsampling rate to 300 results in considerably spread in the points. For $r = 300$, fast solar wind has the highest entropy and lowest FIM in almost all cases. For $B_x$ and $B_y$ MC 1 stands out from the rest of the markers, with higher FIM and lower entropy than the other solar wind time series. Similarly to the Jensen-Shannon plane, the events are placed closer to the corner of the plane for $B_z$ than for the other components.

### 3.2.2 HVG degree distribution

Next, we will consider the Fisher-Shannon information plane when it is formed from the Horizontal Visibility Graph (HVG) degree distribution. We remind that therefore there is no subsampling performed. The entropies have been normalized with a maximal entropy, following Ravetti et al. (2014). The maximal entropy is calculated from a series of fractional Gaussian noise with the same length as the studied solar wind series. In this version of the information plane, $B$ again stands out from the individual magnetic field components. In the figures for $B_x$, $B_y$, and $B_z$ all solar wind markers are placed in a cluster below

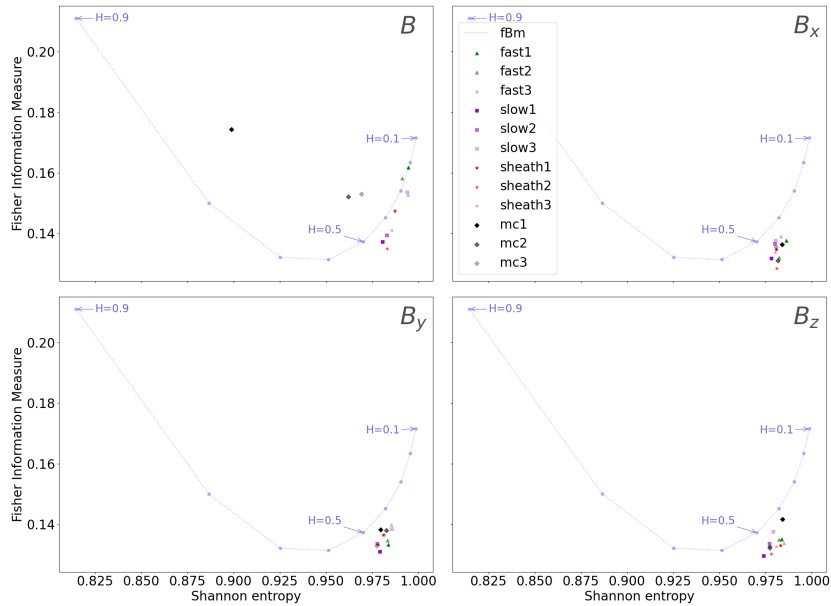

**Figure 5.** The Fisher-Shannon information plane formed from the HVG degree distribution.

the fBm curve. For $B$, the magnetic cloud data is placed away from the curve, and the rest of the solar wind events are located along the curve or under it.

Figure 6 again gives a zoomed-in section of the bottom right corner of the information plane. The highest entropies for $B$ are with the fast solar wind and one of the slow solar wind events (Slow3). The MC events have the lowest entropies. For the magnetic field components, the entropy and FIM values are very similar for all events, and there appears no clear order between different types of solar wind data.

### 3.3 $\lambda$ classification

In addition to forming the information planes, we tested the HVG degree-distribution based classification proposed by Lacasa and Toral (2010). The method is based on the assumption that the tail of the degree distribution follows an exponential rule $P(k) \sim \exp(-\lambda k)$. In our study, the fitting in the degree distribution was done from the fourth to the 14th degree for all distributions. After the 14th degree, some of the distributions began to deviate strongly from the exponential form. The results of the fitting are given in Table 1. The standard deviation error given is obtained as the error from the fitting.

In general, all of the analysed time series are classified as stochastic, as the $\lambda$ values are higher than the limit of $\ln(3/2) \approx 0.405$. For the magnetic field components $B_x$, $B_y$, and $B_z$ there are no clear trends between the different solar wind types in the $\lambda$ values or the sizes of the errors. For $B$, the highest $\lambda$ values are found for the MC time series. The highest value is for the first magnetic cloud, which also consistently had the lowest entropy out of the analysed time series in all of the complexity/information planes. Thus this technique appears to catch the same structure that is detected by the other methods.

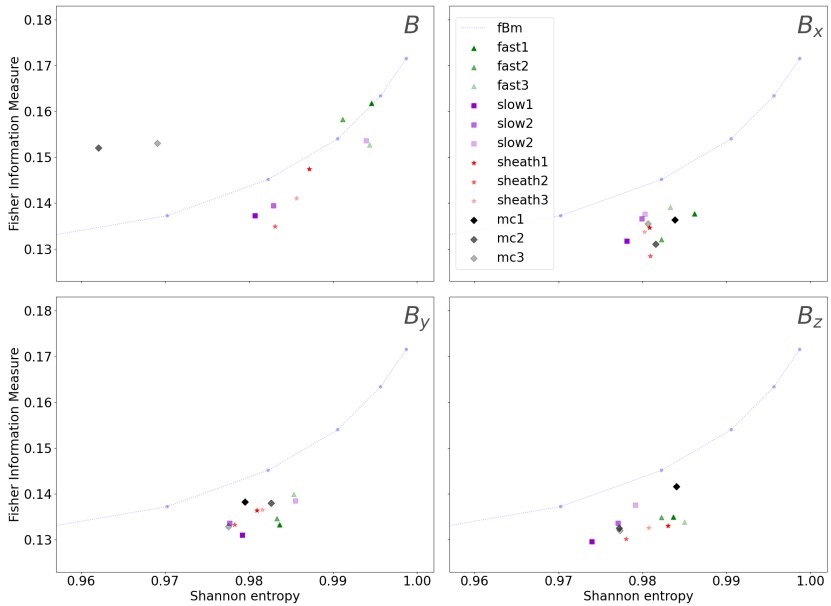

**Figure 6.** A zoomed-in section of the Fisher-Shannon information plane for the HVG degree distribution.

The fast solar wind time series have the lowest $\lambda$ values, closest to the limiting value of $\ln(3/2)$. Lacasa and Toral (2010) suggest that a smaller value in the stochastic region $(> \ln(3/2))$ is an indication of decreasing correlations.

## 4   Discussion

We have applied several methods from information theory and complex networks to study four distinct types of solar wind intervals (fast and slow streams, sheaths and magnetic clouds). In all analysed cases, the most significant differences between the solar wind types occurred when examining the magnetic field magnitude $B$. This finding can likely be attributed to $B$ time series being generally less noisy than the time series for the individual magnetic field components (see the Appendix).

For the approaches using the Jensen-Shannon complexity-entropy plane and the Fisher-Shannon plane with permutation entropy, two values of subsampling rate $r$ were tested. It was found that the higher $r$ value ($r = 300$, corresponding 15 minute time lag between the points used to build the ordinal patterns) causes clearer separation of the solar wind intervals on the planes, with the events spreading out to a larger range of entropies, complexities and FIM values. In particular, for $r = 300$ the fast solar wind events had the highest entropies out of the analysed time series. For $r = 20$ they did not separate from the other investigated solar wind types.

The effect of $r$ is is consistent with the studies by Weck et al. (2015) and Olivier et al. (2019), who studied permutation entropy for fast and slow solar wind $B_x$, and found that a larger $\tau$ ($\tau = r\Delta t$) resulted in larger values of entropy. Olivier et al. (2019) found that when increasing $\tau$, past a value of 180 s the entropy values become stable. Both Weck et al. (2015) and

| sw type, B | $\lambda$ | stde |
|---|---|---|
| fast1 | 0.433 | 0.008 |
| fast2 | 0.444 | 0.01 |
| fast3 | 0.435 | 0.015 |
| slow1 | 0.485 | 0.009 |
| slow2 | 0.498 | 0.009 |
| slow3 | 0.446 | 0.009 |
| sheath1 | 0.48 | 0.012 |
| sheath2 | 0.479 | 0.009 |
| sheath3 | 0.477 | 0.009 |
| MC1 | 0.617 | 0.024 |
| MC2 | 0.525 | 0.014 |
| MC3 | 0.5 | 0.018 |

| sw type, Bx | $\lambda$ | stde |
|---|---|---|
| fast1 | 0.477 | 0.005 |
| fast2 | 0.517 | 0.014 |
| fast3 | 0.486 | 0.008 |
| slow1 | 0.486 | 0.004 |
| slow2 | 0.505 | 0.008 |
| slow3 | 0.486 | 0.007 |
| sheath1 | 0.467 | 0.015 |
| sheath2 | 0.489 | 0.01 |
| sheath3 | 0.469 | 0.02 |
| MC1 | 0.465 | 0.013 |
| MC2 | 0.477 | 0.012 |
| MC3 | 0.508 | 0.012 |

| sw type, By | $\lambda$ | stde |
|---|---|---|
| fast1 | 0.474 | 0.009 |
| fast2 | 0.479 | 0.008 |
| fast3 | 0.475 | 0.005 |
| slow1 | 0.478 | 0.009 |
| slow2 | 0.481 | 0.005 |
| slow3 | 0.464 | 0.01 |
| sheath1 | 0.479 | 0.013 |
| sheath2 | 0.469 | 0.011 |
| sheath3 | 0.47 | 0.008 |
| MC1 | 0.479 | 0.011 |
| MC2 | 0.484 | 0.014 |
| MC3 | 0.476 | 0.005 |

| sw type, Bz | $\lambda$ | stde |
|---|---|---|
| fast1 | 0.486 | 0.006 |
| fast2 | 0.492 | 0.008 |
| fast3 | 0.487 | 0.009 |
| slow1 | 0.479 | 0.005 |
| slow2 | 0.473 | 0.009 |
| slow3 | 0.484 | 0.01 |
| sheath1 | 0.47 | 0.01 |
| sheath2 | 0.506 | 0.009 |
| sheath3 | 0.465 | 0.007 |
| MC1 | 0.44 | 0.014 |
| MC2 | 0.466 | 0.017 |
| MC3 | 0.507 | 0.018 |

**Table 1.** $\lambda$ values for the different solar wind types and components

Olivier et al. (2019) found fast solar wind to have higher entropy than slow solar wind, when studying the $B_x$ component. Looking at Figure 2 we see that for $B_x$ with $r = 300$ the fast solar wind has the highest entropies, with the slow solar wind settling on the fBm curve with lower entropy, in agreement with the studies by Weck et al. (2015) and Olivier et al. (2019). When using $r = 20$ the separation of the two is not as clear, but rather the fast and slow solar wind are closer together on the fBm curve.

Kilpua et al. (2024) also studied the effect of increasing $\tau$ on entropy and Jensen complexity values, analysing sheath regions, SIRs, fast solar wind, slow solar wind, and MCs. They found that increasing $\tau$ resulted in relatively stable values of entropy and complexity for all solar wind types except for MCs, where increasing $\tau$ resulted in smaller entropy values and higher

complexity. In our study, when increasing $\tau$ from 1 minute ($r = 20$) to 15 minutes ($r = 300$) we see change in all solar wind types, in some cases to higher entropies/complexities and in some cases to lower entropies/complexities. The most significant change, however, is in the MCs, which are always moved towards lower entropies and higher complexities, similarly to Kilpua et al. (2024). As was found by Kilpua et al. (2024), these results also refer to the universality of fluctuations at smaller time scales, except in the case highly-ordered magnetic clouds.

Weygand and Kivelson (2019) also used the Jensen-Shannon plane, classifying turbulent intervals and ICME and co-rotating interaction regions (CIRs). They found that turbulent intervals (containing both fast and slow solar wind intervals), when studying magnetic field magnitude, clustered close to the fBm-curve indicating a stochastic nature. This is also the case in our results, though with the higher subsampling rate $r$ some of the slow wind and sheath region intervals move into slightly higher complexity, possibly indicating some chaotic structure (see Figure 1 in Weygand and Kivelson, 2019). Weygand and Kivelson (2019) also analysed the $B_z$ component of ICMEs and CIRs, finding that the analysed events cluster below the fBm curve at high entropies and low complexities. When we analyse the $B_z$ component of MCs, we also see the values settling close to the fBm curve, with the $r = 300$ results closely matching those of Weygand and Kivelson (2019), and $r = 20$ resulting in higher entropy and lower complexity.

A feature of the Jensen-Shannon plane is that Jensen complexity is defined as having small values for either completely ordered or disordered series. This is illustrated for MC1 in $B$ for $r = 300$ (Figure 1), which is placed at zero complexity and entropy on the Jensen-Shannon plane due to its very regular structure. On the Fisher-Shannon plane, this event has FIM of $\sim 0.5$. Furthermore, this MC event also had the highest $\lambda$ value when the exponential fitting introduced by Lacasa and Toral (2010) was applied to the tail of the degree distribution of the HVG that is formed from the time series. A visual inspection of the time series (given in Figure A4) shows that this event clearly has the smoothest signal out of all the analysed events. The two other MC events also stand out from the rest of the solar wind data when $B$ is studied, but not so distinctly.

Using the FIM in combination with permutation entropy instead of Jensen complexity resulted in the solar wind events spreading more along the vertical axis of the information plane. Compared to Jensen complexity, FIM is a more local measure of fluctuations. Even so, the magnetic cloud with the clearest global structure (MC1), which has close to zero Jensen complexity, and a very coherent global structure, has also the highest FIM value out of all of the analysed data. We cannot directly compare our results with the Fisher-Shannon information plane to previous studies, as to our understanding we are the first to utilise it to study solar wind fluctuations. However, the permutation axis is the same as in the Jensen-Shannon plane, and so the points discussed about previous permutation entropy results hold also for these results. A more statistical study of Fisher information and the effect of $\tau$ on it would possibly yield interesting information of the behaviour of solar wind on this plane. In our study we see that increasing $r$ and $\tau$ also had an effect on the FIM value, in most cases resulting in a higher FIM value. In general, on the Fisher-Shannon plane, increasing $r$ led to motion further away from the fBm curve.

In addition to forming the Fisher-Shannon information planes using permutation entropy, we formed them from the HVG degree distribution for each solar wind type. In these figures, the most notable feature is that for all of the magnetic field components ($B_x, B_y, B_z$) the solar wind events cluster below the fBm curve. There is again more deviation in the placements for $B$, with MCs being placed above the fBm curve. The rest of the analysed events are placed closer to the curve, with the

highest entropies found for the fast solar wind and for one of the slow solar wind series. The fact that most of the solar wind series are placed very close to each other indicates that the HVG degree distributions of the series are very similar to each other. Again, we cannot directly compare our results to previous studies using solar wind data. However, to study the technique, Ravetti et al. (2014) analysed several known chaotic and stochastic processes using the Fisher-Shannon plane formed with HVG degree distributions. In their results, the stochastic maps were placed close to the fBm curve, while chaotic processes had higher FIM values and were placed further away from the curve. In our study the MCs, when analysing magnetic field magnitude, were placed in this region close to chaotic processes. When analysing the individual magnetic field components of solar wind, the FIM values are lower than those of the fBm curve. There are no clearly defined regions for stochastic or chaotic values for this plane, but the results for the MC are encouraging, as they are in line with the other methods used in this study that indicate less stochasticity for MCs when analysing magnetic field magnitude.

Lastly, we performed an exponential fit to the tails of the HVG degree distributions. This fitting, following an exponential model, gives an indication of the internal structure of the time series (Lacasa and Toral, 2010). Again, for the magnetic field components there were no clear trends in the values between solar wind types. For $B$, the magnetic clouds had the highest values, indicating a correlated stochastic process. The highest $\lambda$ value was obtained by the magnetic cloud that stood out from the rest of the time series on the information planes. To our understanding, the method has not been previously applied to solar wind time series. However, the results indicate a stochastic nature of solar wind which is in line with previous research. As was mentioned in Section 1, there are some known issues with the method. In our study we perform the exponential fitting from the fourth to the 14th degree, thus leaving out the first degrees of the distribution. By doing this, information about the connections in the network is invariably left out. The technique is perhaps most useful as companion to other methods of complexity analysis, as it does not make full use of the degree distribution.

## 5   Conclusions

Our study shows that different complexity measures gave overall similar results for the analysed time series of solar wind measurements. We analysed four types of solar wind data (fast, slow, sheath regions and magnetic clouds) using the Jensen-Shannon complexity plane and the Fisher-Shannon information plane. Additionally we made use of the horizontal visibility graph (HVG) method in combination with the Fisher-Shannon plane, and via studying the degree distribution of the HVG graphs derived from the time series. All of these methods pointed to the solar wind fluctuations being stochastic for the most part. The degree distribution classification was also in agreement with the other methods. However, as mentioned previously, the technique does not make use of the full degree distribution, and has some known issues.

The most significant finding of our study is that the magnetic cloud data consistently stood out from the other types of solar wind. The analysed magnetic clouds had more global structure and internal cohesion than the other solar wind data types, which is physically consistent with how they are created. A more robust statistical study with a larger sample size could be useful for examining the methods used here in more detail. The Jensen-Shannon complexity plane has already been used within

the solar physics field in several studies, but the Fisher-Shannon information plane in combination with the HVG approach has not been widely used, and could provide interesting insight into the internal structure of solar wind.

*Data availability.* The data and scripts used to produce the files can be accessed via Zenodo (Koikkalainen et al., 2025).

## Appendix A: Solar wind data

*Author contributions.* The research was planned primarily by VK and EK. VK performed the main analysis and wrote main part of the text. EK contributed to the writing of the text. SG and AO also contributed to the writing of the paper, commented the manuscript and provided insight into the methods and results. The Jensen - Shannon complexity algorithm used here was originally coded by AO, VK prepared the 415 other used codes.

*Competing interests.* There are no competing interests

*Acknowledgements.* We acknowledge the Finnish Centre of Excellence in Research of Sustainable Space (Academy of Finland grant number 312390). E.K. acknowledges the ERC under the European Union's Horizon 2020 Research and Innovation Programme Project SolMAG 724391. S.G. is supported by Academy of Finland grants 338486 and 346612 (INERTUM).

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

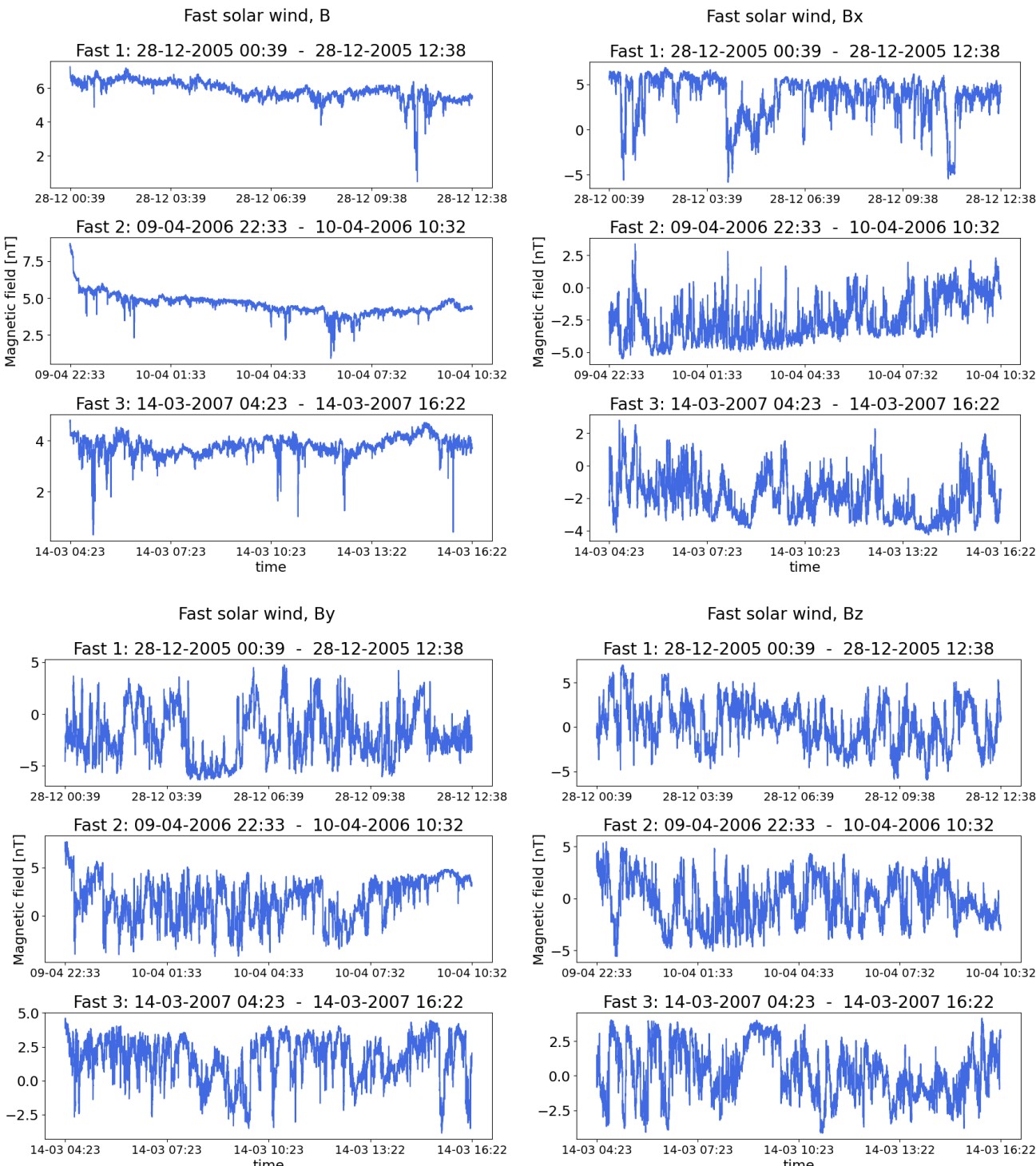

**Figure A1.** The fast solar wind data from WIND.

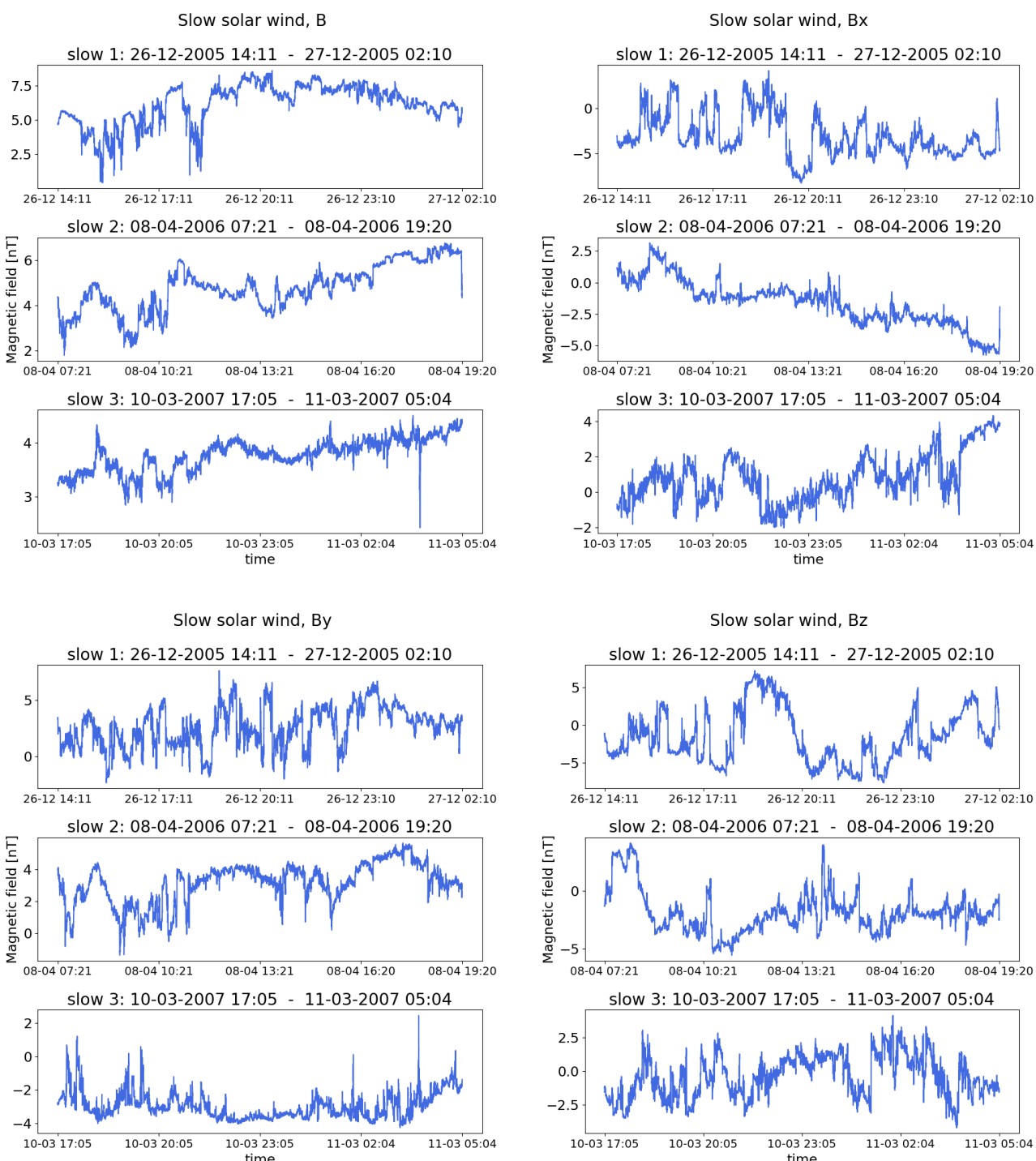

**Figure A2.** The slow solar wind data from WIND.

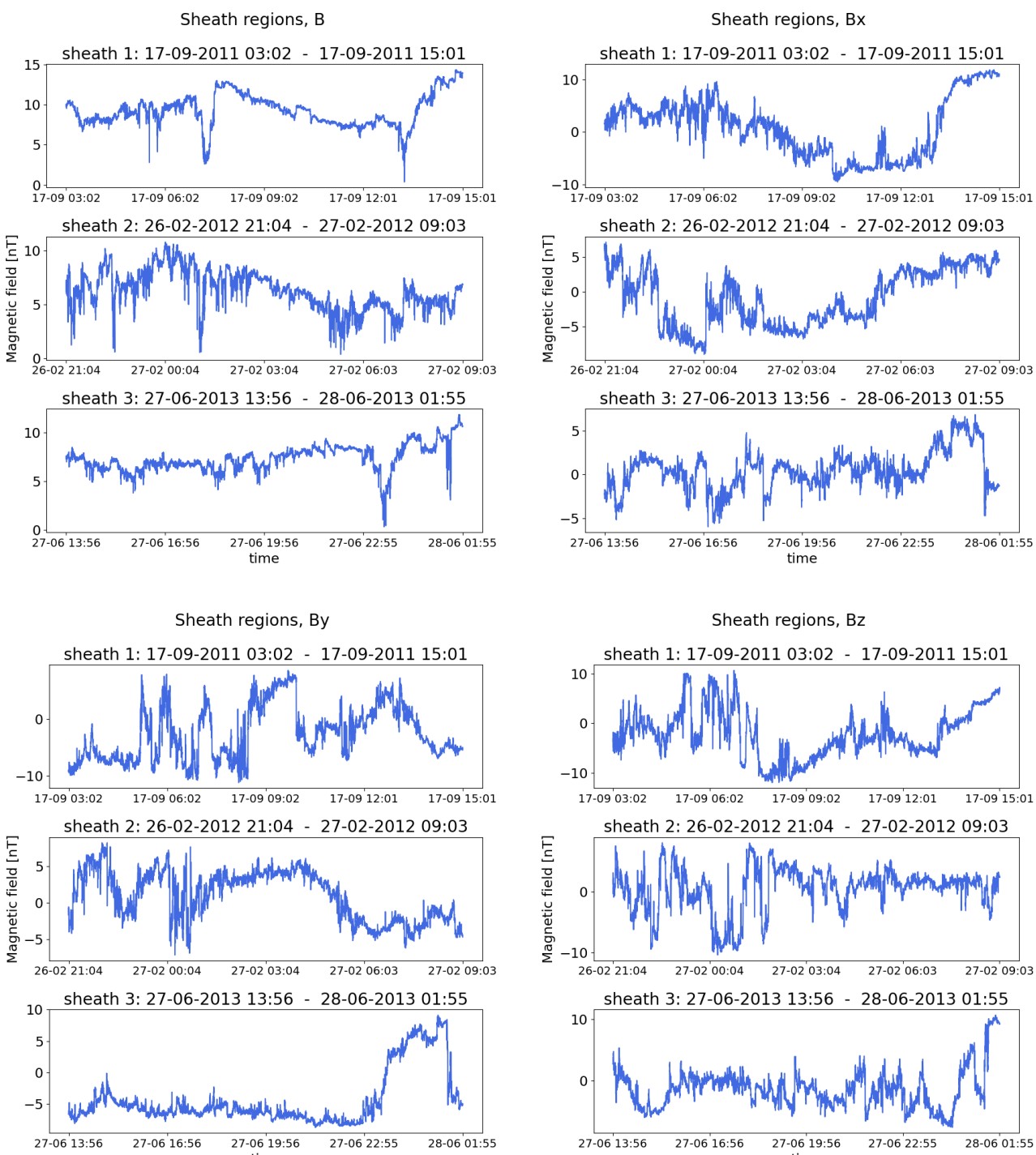

**Figure A3.** The sheath regions from WIND data.

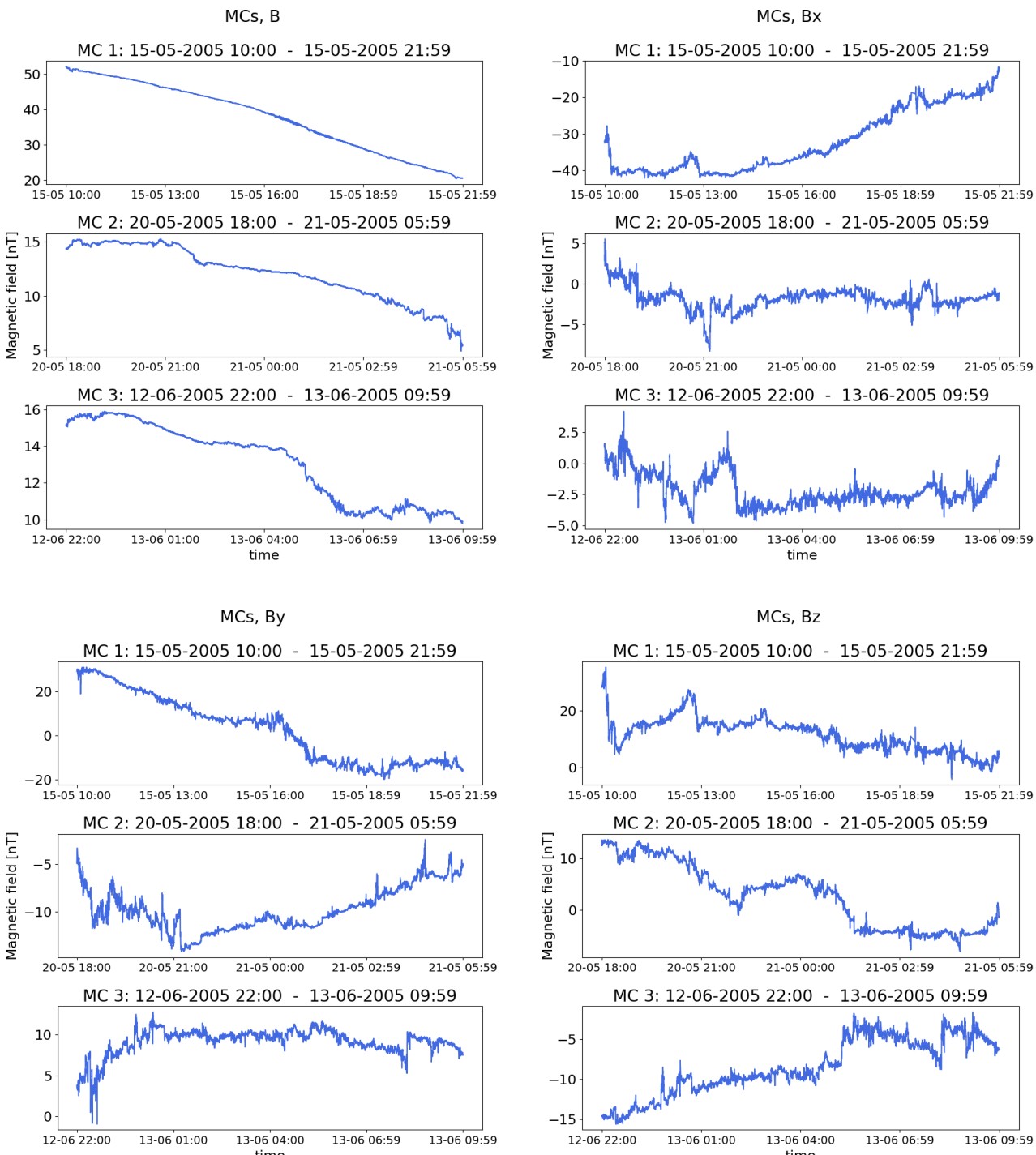

**Figure A4.** The magnetic clouds from WIND data.