# Peer review of "Exploring Complexity Measures for Analysis of Solar Wind Structures and Streams"

_EGUsphere, 2025_

## Referee Comment (RC1)

The authors' application of Jensen-Shannon complexity and Fisher-Shannon information plane to solar wind fluctuations yields interesting and relevant findings for space plasma physics. The methodology is clearly and thoroughly explained, and the results and discussion sections of the manuscript are well-organized and effectively presented. The study is worthy of publication in NPG, with a minor correction.

The authors should cite previous studies that have investigated solar wind time series using entropy and nonlinear dynamics concepts, which have established the stochastic nature of solar wind. Please see my review comments below.

Introduction section

Page 1, Line 20: Please remove the abbreviation 'e.g.' from the citation bracket. Additionally, ensure that all instances of 'e.g.' are removed throughout the entire manuscript.

Page 2, Lines 55-65: While you discuss previous studies that applied Jensen-Shannon complexity analysis to solar wind fluctuations, you omit relevant literature that utilized entropy measures and other nonlinear dynamics tools to investigate solar wind fluctuations. These studies have consistently reported that solar wind exhibits stochastic behavior. Please consider incorporating these references to provide a more comprehensive overview of the field. See the article below for example.

https://doi.org/10.1016/j.asr.2008.12.026

https://doi.org/10.5194/npg-28-257-2021

https://doi.org/10.1029/2018JA025318

https://doi.org/10.1007/s41614-022-00095-z

Page 3, Line 65: "The Jensen-Shannon complexity analysis is only one of a number of methods to investigate the nature of fluctuation". The Jensen-shannon complexity metric is not the only information theory tools that have been applied in space plasma physics. The statement can be corrected as "The Jensen-Shannon complexity metric is also one of the information theory techniques that is useful to investigate the nature of solar wind fluctuation"

Page 3, Line 75-80: Rephrase the statement "The key purpose of the analysis presented in this paper has been to investigate how different complexity measures compared for different solar wind types presented above.

To

"The key purpose of this analysis is to examine how Jensen-Shannon complexity and Fisher-Shannon information plane capture the fluctuation signatures of distinct solar wind structures, including slow streams, fast streams, sheaths, and magnetic clouds."

Data and Methods section

Page 3, line 85: It is better to use "The solar wind data used in this study"

---

## Author Comment (AC1)

**Response to reviewer 1, author comments given in bold text:**

**General comment:**

The authors' application of Jensen-Shannon complexity and Fisher-Shannon information plane to solar wind fluctuations yields interesting and relevant findings for space plasma physics. The methodology is clearly and thoroughly explained, and the results and discussion sections of the manuscript are well-organized and effectively presented. The study is worthy of publication in NPG, with a minor correction.

The authors should cite previous studies that have investigated solar wind time series using entropy and nonlinear dynamics concepts, which have established the stochastic nature of solar wind. Please see my review comments below.

**-The authors thank the referee for the comments and suggestions, which will improve the manuscript quality. The additional references are valuable and will be added to the paper to have a more comprehensive view of the previous literature.**

**Review comments:**
Introduction section
Page 1, Line 20: Please remove the abbreviation 'e.g.' from the citation bracket. Additionally, ensure that all instances of 'e.g.' are removed throughout the entire manuscript.

**-The abbreviations can be removed.**

Page 2, Lines 55-65: While you discuss previous studies that applied Jensen-Shannon complexity analysis to solar wind fluctuations, you omit relevant literature that utilized entropy measures and other nonlinear dynamics tools to investigate solar wind fluctuations. These studies have consistently reported that solar wind exhibits stochastic behavior. Please consider incorporating these references to provide a more comprehensive overview of the field. See the article below for example.

https://doi.org/10.1016/j.asr.2008.12.026

https://doi.org/10.5194/npg-28-257-2021
https://doi.org/10.1029/2018JA025318

https://doi.org/10.1007/s41614-022-00095-z

**-These references appear to be relevant for the paper, we will incorporate them into the text.**

Page 3, Line 65: "The Jensen-Shannon complexity analysis is only one of a number of methods to investigate the nature of fluctuation".  The Jensen-Shannon complexity metric is not the only information theory tools that have been applied in space plasma physics. The statement can be corrected as "The Jensen-Shannon complexity metric is also one of the information theory techniques that is useful to investigate the nature of solar wind fluctuation"

**-This is a good point, we will amend the text to be clearer  on this point.**

Page 3, Line 75-80: Rephrase the statement "The key purpose of the analysis presented in this paper has been to investigate how different complexity measures compared for different solar wind types presented above.

                    To

"The key purpose of this analysis is to examine how Jensen-Shannon complexity and Fisher-Shannon information plane capture the fluctuation signatures of distinct solar wind structures, including slow streams, fast streams, sheaths, and magnetic clouds."

**-This way of formatting the statemen is more clear, we will change the paper to reflect this. In addition to the Jensen-Shannon and Fisher-Shannon planes, we have the HVG lambda-analysis as well.**

Data and Methods section
Page 3, line 85: It is better to use "The solar wind data used in this study"

**-We agree that this is a clearer way of describing the data and will make the suggested change.**

**Response to reviewer 2, author comments given in bold text:**

**General  comment:**

This is an innovative and original study, contributing new findings to the existing knowledge of dynamical complexity in the solar-terrestrial system. The authors apply information

theory measures to investigate the complex character of the dynamics of solar wind. The manuscript is well-written and deserves to be published in NPG following a minor revision. These are my remarks:

There are two recent review articles highlighting the importance of information theory for the study of coupling processes in the solar-terrestrial system (Balasis et al., 2023; McGranaghan, 2024). Please consider mention these pertinent review articles either in Introduction and / or Discussion.

**-The authors thank the reviewer for the valuable comments and the thorough reading of the manuscript. We have gone  through the comments carefully and will make the changes accordingly.**

**Review comments:**

L. 22: The solar wind exhibits large-scale organisation,

I think that the word 'organisation' here could create some mild confusion to the reader, because the expression 'large-scale organisation' is (a) often used in the field of business and finance in a different context, and/or (b) could be linked by a reader to systems exhibiting self-organized criticality (SOC) etc. Maybe 'structure' instead of 'organisation' is a more suitable word here.

**-We agree, structure would be a suitable word here to avoid potential confusion.**

L. 34-35: ICME sheaths as compressed structures more resemble SIRs

Do you mean 'may resemble SIRs' or something else?

 **-We are making a comparison between SIRs and sheaths, we will change the 'more resemble SIRs' to 'are more similar to' to make this clearer.**

L. 56: The nature of solar wind fluctuations can be assessed using Jensen-Shannon complexity analysis

Please elaborate in the text on the following interconnected points:

1. What is the rationale of using complexity analysis for assessing the nature of solar wind fluctuations?

2. If you establish the reasoning for point 1, why then to use the specific analysis? Which is the benefit from performing time series analysis through this measure given the plethora of the available information theory techniques?

**-We will further elaborate on this in the revised text.**

**Question 1: The rationale of using complexity analysis is to find more about the underlying processes that might be responsible for the creation of fluctuations in the different types of solar wind and to separate different intervals. Using complexity analysis can be beneficial for finding, for example, if the process is chaotic or stochastic. It has generally been found that solar wind fluctuations are stochastic, and we wanted to see if this is the case in different types of solar wind (fast, slow, sheaths, MCs). Variation within certain driver types in information complexity measures can also give insight to their overall structural complexities (e.g. in the case of magnetic clouds). There have been previous studies into varying solar wind types and complexity analysis, but not with the set of methods that we used.**

**- Question 2: The idea behind choosing these methods was to find a new perspective into analysis of solar wind fluctuation analysis, while expanding the use of methods that have been used in some way in solar/space physics. There has been some use of the HVG method in the field, and we wanted to test the method for solar wind data. To our understanding, the HVG method in combination with entropy and the Fisher-Shannon information plane is new to our field. We wanted to test this while comparing the results to a similar method that has been used previously in several studies, i.e. the Jensen-Shannon complexity-entropy plane. We believe the two methods complement each other.**

L. 65: magnetic clouds clearly exhibited the lowest entropies and highest complexities

I am not sure that lower entropy means higher complexity for a system. I would say the opposite: lower entropy means higher order and therefore less complexity for a system. Could you please explain this point?

**-Indeed, in the complexity-entropy figures presented in the discussed study, high values of entropy indicate high disorder and vice versa. As magnetic clouds have a coherent structure, their entropy value is low. However, the MC structure is captured in the Jensen-Shannon complexity measure, which gives low values for both high order and high disorder, and high values for cases in between. The MCs thus have enough structure to have low entropy, but enough variation in the structure that their**

**Jensen-Shannon complexity is higher. We will clarify this in the paper by mentioning that it was specifically the Jensen-Shannon complexity that was highest for the magnetic clouds.**

L. 74: There are a few earlier as well as more recent applications of Fisher information in the context of geophysics / geomagnetism and space physics / space weather (see for instance, Balasis et al., 2016, 2023, respectively).

**-Thank you for these suggestions, we had not come across these studies. We will review them and add to the text accordingly.**

L. 86-87: Three intervals of data were considered for each solar wind type, each consisting of 12 hours of measurements.

Please consider changing this point a bit by adding the specific times of the three intervals and mentioning the number of solar wind type to make it more clear for the reader:

Three time intervals (1) from … to …, 2) from … to …, 3) from … to …) of data were considered for each of the four solar wind types, each consisting of 12 hours of measurements.

**-This will indeed make this point clearer, we will add the interval times to the text.**

L. 295: overall, what is clearly absent/missing in the Discussion is the (expected) comparison to other/similar space physics studies using information theory measures. Please provide in this section such a useful comparison.

**-Yes, a comparison to previous research would be beneficial, we will add this to the section.**

L. 347: The analysed magnetic clouds had more internal structure than the other solar wind data

Could you please elaborate a bit on this point? Does more internal structure mean more order / lower entropy / lower complexity or vice versa in your perspective?

**-We will clarify this point further by mentioning which of the methods this is related to and how. It would be better to refer to this as the global rather than the "internal" structure of the time series, the text will be changed to reflect this.**

References

Balasis, G., S. M. Potirakis, and M. Mandea (2016), Investigating Dynamical Complexity of Geomagnetic Jerks using Various Entropy Measures, Front. Earth Sci., 4:71, doi:10.3389/feart.2016.00071.

Balasis, G.; Boutsi, A.Z.; Papadimitriou, C.; Potirakis, S.M.; Pitsis, V.; Daglis, I.A.; Anastasiadis, A.; Giannakis, O. Investigation of dynamical complexity in Swarm-derived geomagnetic activity indices using information theory. Atmosphere 2023, 14, 890. https://doi.org/10.3390/atmos14050890

Balasis, G., Balikhin, M.A., Chapman, S.C. et al. Complex Systems Methods Characterizing Nonlinear Processes in the Near-Earth Electromagnetic Environment: Recent Advances and Open Challenges. Space Sci Rev 219, 38 (2023). https://doi.org/10.1007/s11214-023-00979-7

McGranaghan, R.M. Complexity Heliophysics: A Lived and Living History of Systems and Complexity Science in Heliophysics. Space Sci. Rev. 2024, 220, 52.

**-Thank you for these references, we will review them and add to the text.**

---

## Author Response (AR1)

**General comments:**

**We thank the referees for their comments, and have made changes to the manuscript according to them. We feel the quality of the manuscript has increased as a result of the suggestions. The most significant change has been to the Discussion, which has been reworked to include more comparisons to previous studies. In addition to that, some small phrasing issues have been clarified throughout the text. There are now more comments about the use of complexity analysis on studying fluctuations in the Introduction. We also changed how the date-labels for the solar wind time series in the Appendix are plotted to make the figures more clear. The Zenodo Dataset has been updated accordingly, and the citation pointing to it has the new DOI to access the new version of the codes. This change was only to the labels of the time series figures, and did not affect the results.**

**Response to reviewer 1, author comments given in bold text:**

The authors' application of Jensen-Shannon complexity and Fisher-Shannon information plane to solar wind fluctuations yields interesting and relevant findings for space plasma physics. The methodology is clearly and thoroughly explained, and the results and discussion sections of the manuscript are well-organized and effectively presented. The study is worthy of publication in NPG, with a minor correction.

The authors should cite previous studies that have investigated solar wind time series using entropy and nonlinear dynamics concepts, which have established the stochastic nature of solar wind. Please see my review comments below.

**-The authors thank the referee for the comments and suggestions, the manuscript has been improved by these comments. The suggested references are relevant, and were added to the introduction of the manuscript.**

Review comments:
Introduction section
Page 1, Line 20: Please remove the abbreviation 'e.g.' from the citation bracket. Additionally, ensure that all instances of 'e.g.' are removed throughout the entire manuscript.

**-The abbreviations were removed.**

Page 2, Lines 55-65: While you discuss previous studies that applied Jensen-Shannon complexity analysis to solar wind fluctuations, you omit relevant literature that utilized entropy measures and other nonlinear dynamics tools to investigate solar wind fluctuations. These studies have consistently reported that solar wind exhibits stochastic behavior. Please consider incorporating these references to provide a more comprehensive overview of the field. See the article below for example.

https://doi.org/10.1016/j.asr.2008.12.026

https://doi.org/10.5194/npg-28-257-2021
https://doi.org/10.1029/2018JA025318

https://doi.org/10.1007/s41614-022-00095-z

**-Thank you for these references, they were added to the text**

Page 3, Line 65: "The Jensen-Shannon complexity analysis is only one of a number of methods to investigate the nature of fluctuation".  The Jensen-Shannon complexity metric is not the only information theory tools that have been applied in space plasma physics. The statement can be corrected as "The Jensen-Shannon complexity metric is also one of the information theory techniques that is useful to investigate the nature of solar wind fluctuation"

**-This was indeed perhaps a bit confusing. The meaning of the sentence is to point out that there are other methods besides the Jensen-Shannon complexity. We changed the wording to be more clear on this point, the wording is now:  "The Jensen-Shannon complexity analysis is one of a number of methods to investigate the nature of fluctuations."**

Page 3, Line 75-80: Rephrase the statement "The key purpose of the analysis presented in this paper has been to investigate how different complexity measures compared for different solar wind types presented above.
                        To
"The key purpose of this analysis is to examine how Jensen-Shannon complexity and Fisher-Shannon information plane capture the fluctuation signatures of distinct solar wind structures, including slow streams, fast streams, sheaths, and magnetic clouds."

**-We changed the statement to be "The key purpose of this analysis is to examine how the Jensen-Shannon complexity, the Fisher-Shannon information plane, and HVG-**

**analysis capture the fluctuation signatures of distinct solar wind structures". The solar wind types are introduced very shortly afterwards.**

Data and Methods section
Page 3, line 85: It is better to use "The solar wind data used in this study"

**-We made the change to this phrasing.**

**Response to reviewer 2, author comments given in bold text:**

This is an innovative and original study, contributing new findings to the existing knowledge of dynamical complexity in the solar-terrestrial system. The authors apply information theory measures to investigate the complex character of the dynamics of solar wind. The manuscript is well-written and deserves to be published in NPG following a minor revision. These are my remarks:

There are two recent review articles highlighting the importance of information theory for the study of coupling processes in the solar-terrestrial system (Balasis et al., 2023; McGranaghan, 2024). Please consider mention these pertinent review articles either in Introduction and / or Discussion.

**-The authors thank the reviewer for the valuable comments and the thorough reading of the manuscript. We have gone through the comments carefully and made the changes.**

**Review comments:**

L. 22: The solar wind exhibits large-scale organisation,

I think that the word 'organisation' here could create some mild confusion to the reader, because the expression 'large-scale organisation' is (a) often used in the field of business and finance in a different context, and/or (b) could be linked by a reader to systems exhibiting self-organized criticality (SOC) etc. Maybe 'structure' instead of 'organisation' is a more suitable word here.

**-This is indeed a potential source of confusion, especially related to the second point (b). We changed the word to "structure" to avoid any confusion.**

L. 34-35: ICME sheaths as compressed structures more resemble SIRs

Do you mean 'may resemble SIRs' or something else?

**-We are making a comparison between SIRs and sheaths, the wording was changed to "In contrast to MCs, ICME sheaths, being compressive structures, are more similar to SIRs in their solar wind properties, exhibiting large-amplitude magnetic field variations and relatively high densities and temperatures."**

L. 56: The nature of solar wind fluctuations can be assessed using Jensen-Shannon complexity analysis

Please elaborate in the text on the following interconnected points:

1. What is the rationale of using complexity analysis for assessing the nature of solar wind fluctuations?

2. If you establish the reasoning for point 1, why then to use the specific analysis? Which is the benefit from performing time series analysis through this measure given the plethora of the available information theory techniques?

**-We addressed the questions in the following paragraph, which was added to the introduction:**

**"In this study we apply complexity analysis to study fluctuations in the solar wind, which offers a complementary approach to more traditional analysis techniques. Using complexity analysis we can explore phenomena such as cross-scale effects, emergence, and self-organising behaviour (McGranaghan, 2024). This is particularly relevant to the study of the solar wind, where a plethora of fundamental plasma processes are in action. These processes cause structures from small-scale turbulent fluctuations to large-scale phenomena such as ICMEs. %plethora of effects cause fluctuations at varying scales, from turbulence to ICMEs. While complexity science or information theory may not directly explain the underlying physical processes of the analysed systems, they can provide valuable insights into patterns and structures in solar wind time series, help to identify the combined effects of interacting subsystems, and differentiate between solar wind structures of different origin ( Kilpua et al., 2024). Our aim is to explore techniques that are new to solar wind studies (HVG analysis and the Fisher-Shannon information plane) in combination with a technique that has been used previously in the field, Jensen-Shannon complexity.**

**These methods, which will be introduced in the next paragraphs, are complementary to each other."**

L. 65: magnetic clouds clearly exhibited the lowest entropies and highest complexities

I am not sure that lower entropy means higher complexity for a system. I would say the opposite: lower entropy means higher order and therefore less complexity for a system. Could you please explain this point?

**-Indeed, in the complexity-entropy figures presented in the discussed study, high values of entropy indicate high disorder and vice versa. As magnetic clouds have a coherent structure, their entropy value is low. However, the MC structure is captured in the Jensen-Shannon complexity measure, which gives low values for both high order and high disorder, and high values for cases in between. The MCs thus have enough structure to have low entropy, but enough variation in the structure that their Jensen-Shannon complexity is higher. We will clarify this in the paper by mentioning that it was specifically the Jensen-Shannon complexity that was highest for the magnetic clouds.**

L. 74: There are a few earlier as well as more recent applications of Fisher information in the context of geophysics / geomagnetism and space physics / space weather (see for instance, Balasis et al., 2016, 2023, respectively).

**-Thank you for introducing us to these studies, they are interesting and show the Fisher information has previously been used in the context of space physics. We added the references to the introduction.**

L. 86-87: Three intervals of data were considered for each solar wind type, each consisting of 12 hours of measurements.

Please consider changing this point a bit by adding the specific times of the three intervals and mentioning the number of solar wind type to make it more clear for the reader:

Three time intervals (1) from … to …, 2) from … to …, 3) from … to …) of data were considered for each of the four solar wind types, each consisting of 12 hours of measurements.

**-The time intervals were added to the text, this was a good point.**

L. 295: overall, what is clearly absent/missing in the Discussion is the (expected) comparison to other/similar space physics studies using information theory measures. Please provide in this section such a useful comparison.

**-Thank you, this is a good point that made the discussion better. We re-structured the section to include more comparisons to previous studies, especially for the Jensen-Shannon plane, where there are several relevant previous studies that our results can be compared to. Please refer to the manuscript for the revised discussion.**

L. 347: The analysed magnetic clouds had more internal structure than the other solar wind data

Could you please elaborate a bit on this point? Does more internal structure mean more order / lower entropy / lower complexity or vice versa in your perspective?

**-The phrasing of "internal structure" was not quite correct here, we changed it to "global structure", as this point was about the overall coherence/lack of fluctuations.**

References

Balasis, G., S. M. Potirakis, and M. Mandea (2016), Investigating Dynamical Complexity of Geomagnetic Jerks using Various Entropy Measures, Front. Earth Sci., 4:71, doi:10.3389/feart.2016.00071.

Balasis, G.; Boutsi, A.Z.; Papadimitriou, C.; Potirakis, S.M.; Pitsis, V.; Daglis, I.A.; Anastasiadis, A.; Giannakis, O. Investigation of dynamical complexity in Swarm-derived geomagnetic activity indices using information theory. Atmosphere 2023, 14, 890. https://doi.org/10.3390/atmos1405089

Balasis, G., Balikhin, M.A., Chapman, S.C. et al. Complex Systems Methods Characterizing Nonlinear Processes in the Near-Earth Electromagnetic Environment: Recent Advances and Open Challenges. Space Sci Rev 219, 38 (2023). https://doi.org/10.1007/s11214-023-00979-7

McGranaghan, R.M. Complexity Heliophysics: A Lived and Living History of Systems and Complexity Science in Heliophysics. Space Sci. Rev. 2024, 220, 52.

**-Thank you for these references, especially the review-articles are very good resources. We added these to the manuscript.**